# Risk factors for asthma among schoolchildren who participated in a case-control study in urban Uganda

Harriet Mpairwe[1]*, Milly Namutebi[1], Gyaviira Nkurunungi[1], Pius Tumwesige[1], Irene Nambuya[1], Mike Mukasa[1], Caroline Onen[1], Marble Nnaluwooza[1], Barbara Apule[1], Tonny Katongole[1], Gloria Oduru[1], Joseph Kahwa[1], Emily L Webb[2], Lawrence Lubyayi[1], Neil Pearce[2], Alison M Elliott[1,2]

[1]Medical Research Council/Uganda Virus Research Institute and London School of Hygiene and Tropical Medicine Uganda Research Unit, Entebbe, Uganda; [2]London School of Hygiene and Tropical Medicine, London, United Kingdom

**Abstract** Data on asthma aetiology in Africa are scarce. We investigated the risk factors for asthma among schoolchildren (5–17 years) in urban Uganda. We conducted a case-control study, among 555 cases and 1115 controls. Asthma was diagnosed by study clinicians. The main risk factors for asthma were tertiary education for fathers (adjusted OR (95% CI); 2.32 (1.71–3.16)) and mothers (1.85 (1.38–2.48)); area of residence at birth, with children born in a small town or in the city having an increased asthma risk compared to schoolchildren born in rural areas (2.16 (1.60–2.92)) and (2.79 (1.79–4.35)), respectively; father's and mother's history of asthma; children's own allergic conditions; atopy; and cooking on gas/electricity. In conclusion, asthma was associated with a strong rural-town-city risk gradient, higher parental socio-economic status and urbanicity. This work provides the basis for future studies to identify specific environmental/lifestyle factors responsible for increasing asthma risk among children in urban areas in LMICs.

*For correspondence:
Harriet.Mpairwe@mrcuganda.org

**Competing interests:** The authors declare that no competing interests exist.

## Introduction

Asthma is estimated to affect more than 235 million people globally, and is the most common non-communicable condition among children (*World Health Organisation, 2017*). In Africa, the prevalence of asthma appears to be increasing (*Asher et al., 2006*; *Addo-Yobo et al., 2007*; *van Gemert et al., 2011*; *Zar et al., 2007*; *Lawson et al., 2017*), particularly in urban areas (*Addo-Yobo et al., 2007*; *Morgan et al., 2018*), but the causes of this increase are not fully understood. Moreover, asthma has various phenotypes which may have different aetiologies (*Lötvall et al., 2011*). Asthma risk factors appear to vary internationally, and to differ between high-income countries (HICs) and low-and-middle income countries (LMICs) (*Douwes and Pearce, 2002*).

There is little previously published data on asthma risk factors from Africa. The few studies reported have suggested that current residence in urban areas is associated with a higher risk of asthma than rural residence in Africa (*Botha et al., 2019*; *Addo-Yobo et al., 2001*) and other LMICs (*Robinson et al., 2011*; *Gaviola et al., 2016*). The association between helminth infections and asthma has been investigated in Africa and other LMICs, but the findings have been inconsistent across studies (*Leonardi-Bee et al., 2006*; *Mpairwe and Amoah, 2019*). Other risk factors for asthma in Africa and other LMICs, similar to those in HICs, include maternal smoking (*Ayuk et al., 2018*; *Arrais et al., 2019*), maternal history of asthma (*Nantanda et al., 2013*), childhood atopic sensitisation (*Addo-Yobo et al., 2001*; *Nyembue et al., 2012*) and history of allergy (*Nantanda et al., 2013*; *Mehanna et al., 2018*). Previous reports suggest no association between biomass fuels and asthma risk (*Thacher et al., 2013*; *Oluwole et al., 2017a*), but increased asthma

**eLife digest** Asthma is a chronic disease of the airways that leads to breathing difficulty and sometimes death: the condition affects about 235 million people worldwide, especially children. Scientists still do not know exactly what causes asthma, but studies in Europe and North America suggest that individuals born or raised in rural areas are less likely to be affected. However, few studies have examined asthma in African countries, where urbanization is often quickly increasing. Examining the factors associated with the disease as more people move to cities may provide new clues about how asthma emerges, and how to prevent it.

To this end, Mpairwe et al. conducted a study with over 1,670 schoolchildren in Uganda. Those born or raised in rural areas were least likely to have asthma, but the risk doubled among children from small towns, and tripled in those born or who grew up in the city. Children whose parents had a higher education and socioeconomic status had the highest asthma risk, but more work is required to understand why this is the case.

The study by Mpairwe et al. is the first step towards identifying environmental and lifestyle factors associated with increased asthma risk in Africa. Further studies may help scientists to understand how beginning life in a more urban area plays a role in the development of the disease.

symptoms (*Oluwole et al., 2017a*; *Oluwole et al., 2017b*). Unlike in HICs, higher parental education and socioeconomic status has been associated with asthma among children in Africa (*Addo-Yobo et al., 2007*; *Nantanda et al., 2013*; *Wolff et al., 2012*).

We undertook a case-control study among schoolchildren in an urban area in Uganda, to investigate the main risk factors for asthma and the patterns of allergic sensitisation.

## Results

### Reference characteristics of participating schools and participant flow

We enrolled participants from 55 schools (32 primary, 23 secondary). Of the 6385 children initially identified from the pre-screening exercise, we were unable to contact 4550 parents/guardians in time for them to attend the parents' meeting; most of these children were in the boarding section (*Figure 1*). Of the 1835 who attended the meeting, 97% provided written informed consent for their child to participate in the study. We screened 1779 participants and of these, 77 who had initially reported breathing problems either did not have an asthma diagnosis or did not have asthma symptoms in the last 12 months and were excluded. We enrolled 562 children with and 1140 without asthma, but excluded thirty-two with incomplete data (*Figure 1*). At enrolment, 477 asthma cases successfully performed the spirometry, and only three of these had $FEV_1$ values less than 80% of predicted values.

### Early life risk factors for asthma

Participants had mean age 11 years (range 5–17 years); children with asthma were slightly older, and more likely to have parents with a tertiary education and a reported history of asthma (*Table 1*). Compared to children born in rural Uganda, children born in any town in Uganda or in the city had an increased risk of asthma [adjusted OR (95% confidence interval (CI)) 2.16 (1.60–2.92)] and [2.79 (1.79–4.35)], respectively. The same pattern was observed for the area where the child spent most of their early life (0–5 years) (*Table 1*). There were no differences in reported exposure to farm animals, or to cigarette smoke during pregnancy. Children with asthma were less likely to have a BCG scar [0.67 (0.51–0.89)] (*Table 1*), but the TST response (induration ≥10 mm) at enrolment was similar between cases (15.6%) and controls (14.4%) [1.03 (0.67–1.58)]. There was no statistical evidence of interaction between parental education and the children's area of residence in early life, nor interaction between age and any of the asthma risk factors.

### Current features of asthma cases versus controls

Asthma cases were more likely to report a high frequency of 'trucks passing on the street near their home' [2.28 (1.52–3.43)]; to come from homes that used electricity/gas for indoor cooking [1.58

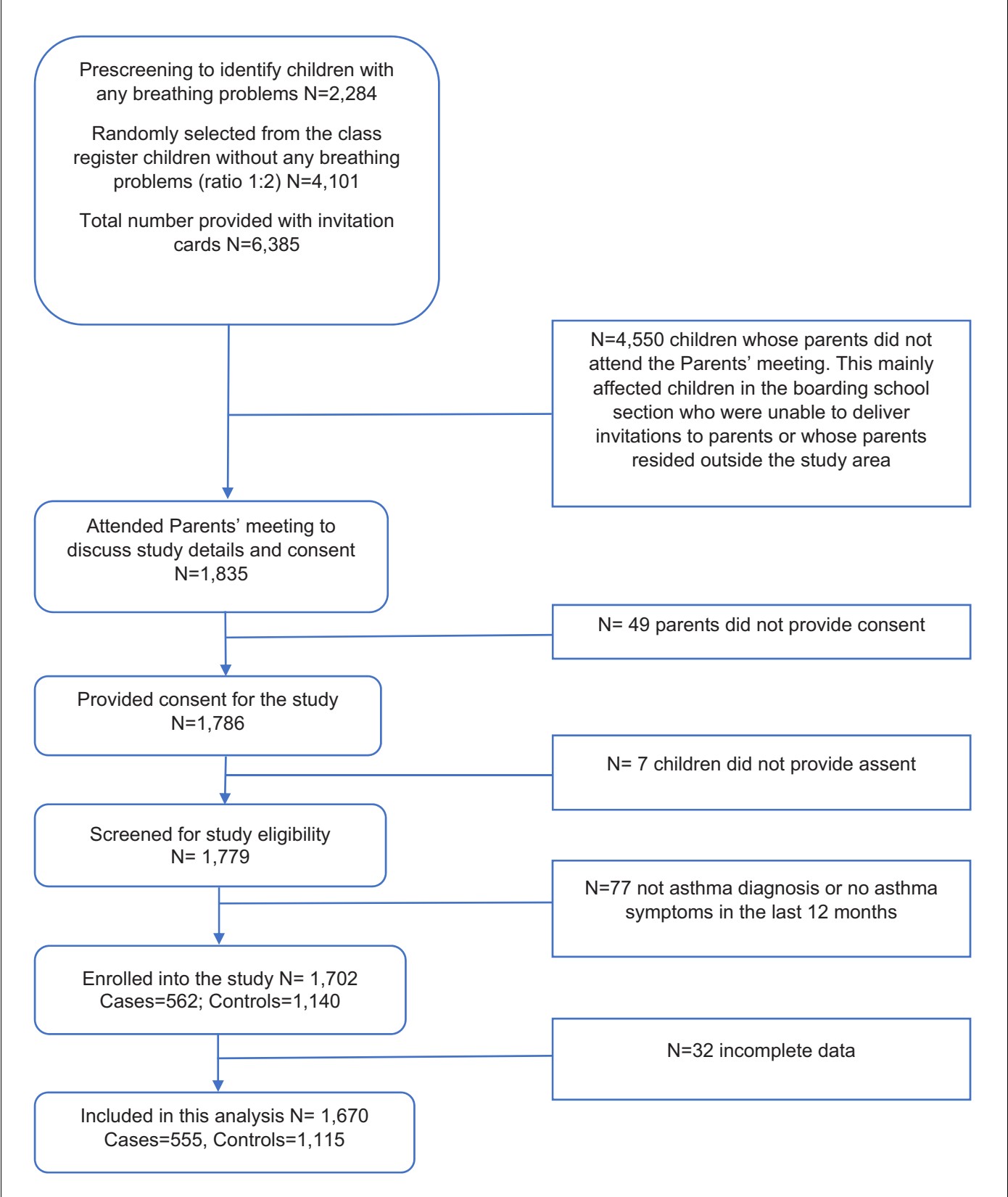

**Figure 1.** Participant flow diagram for an asthma case-control study, conducted among 1670 schoolchildren aged 5–17 years, between 2015 and 2017 in Uganda.

**Table 1.** Early life risk factors for asthma among schoolchildren enrolled in a case-control study between 2015–17 (N = 1,670)

| Risk factors | Asthma cases N = 555 | Non-asthma controls N = 1115 | Adj. OR (95% CI)[*] |
|---|---|---|---|
| Age, years, Mean (Range) | 11.42 (3.26) | 10.98 (3.05) | 1.07 (1.03–1.11) |
| Sex: Girls (933) | 294 (53.0) | 639 (57.3) | 0.79 (0.63–0.99) |
| Mother's history of asthma [m = 164] | | | |
| Yes (102) | 61 (12.0) | 41 (4.1) | 3.05 (1.97–4.71) |
| Father's history of asthma [m = 175] | | | |
| Yes (85) | 65 (12.8) | 20 (2.0) | 6.64 (3.90–11.30) |
| Fathers' highest education level | | | |
| None/Primary (437) | 99 (17.8) | 338 (30.3) | 1 |
| Secondary (594) | 193 (34.8) | 401 (36.0) | 1.61 (1.20–2.16) |
| Tertiary (639) | 263 (47.4) | 376 (33.7) | 2.32 (1.71–3.16)[†] |
| Mothers' highest education attained [m = 4] | | | |
| None/Primary (577) | 158 (28.6) | 419 (37.6) | 1 |
| Secondary (601) | 185 (33.4) | 416 (37.4) | 1.11 (0.85–1.44) |
| Tertiary (488) | 210 (38.0) | 278 (25.0) | 1.85 (1.38–2.48)[†] |
| Child's residence at birth | | | |
| Rural (352) | 74 (13.3) | 278 (24.9) | 1 |
| Town (1,169) | 412 (74.2) | 757 (67.9) | 2.16 (1.60–2.92) |
| City (149) | 69 (12.5) | 80 (7.2) | 2.79 (1.79–4.35)[†] |
| Child's residence for most of 0–5 years | | | |
| Rural (337) | 72 (13.0) | 265 (23.8) | 1 |
| Town (1,247) | 434 (78.2) | 813 (72.9) | 1.96 (1.44–2.66) |
| City (86) | 49 (8.8) | 37 (3.3) | 3.74 (2.19–6.39)[†] |
| Maternal regular contact with farm animals during pregnancy [m = 252] | | | |
| Yes (500) | 152 (30.2) | 348 (38.0) | 0.82 (0.64–1.06) |
| Maternal cigarette smoking during pregnancy [m = 302] | | | |
| Yes (36) | 13 (2.7) | 23 (2.6) | 1.07 (0.51–2.22) |
| Breast feeding duration [m = 402] | | | |
| ≤1 year (328) | 114 (25.3) | 214 (26.2) | 1 |
| >1 year (940) | 337 (74.7) | 603 (73.8) | 1.12 (0.85–1.49) |
| Allergic rhinitis ever | | | |
| Yes (862) | 429 (77.3) | 433 (38.8) | 5.53 (4.31–7.09) |
| Allergic conjunctivitis ever | | | |
| Yes (766) | 361 (65.0) | 405 (36.3) | 3.12 (2.50–3.90) |
| Eczema ever | | | |
| Yes (264) | 126 (22.7) | 138 (12.4) | 2.08 (1.56–2.77) |
| Urticarial rash ever | | | |
| Yes (569) | 222 (40.0) | 347 (31.1) | 1.42 (1.13–1.79) |
| Presence of a BCG scar on child's arm at enrolment [m = 83] | | | |
| Yes (1,315) | 413 (78.4) | 902 (85.1) | 0.67 (0.51–0.89) |

Adj. OR = adjusted odds ratio; CI = confidence interval; N = number; m = missing; number (%) in 2nd and 3rd column unless indicated otherwise.

[*]adjusted for child's age, sex, residence at birth and father's education level.

[†]Test for trend p-value<0.0001. For parental history of allergy, 6.6% mothers and 3.4% fathers responded by telephone.

(1.16–2.17)]; and to report having used de-worming medication more than twice in the last 12 months [2.18 (1.62–2.93)] than controls (*Table 2*). There was weak evidence for an inverse association between asthma and infection with any helminths species [0.75 (0.53–1.07)] (*Table 2*); overall this was not statistically significant as shown by the fact that the confidence interval included the null value of 1, but it was significant for *T. trichiura* [0.33 (0.13–0.89)] (*Supplementary file 1a*). Children with asthma were more likely to report a history of allergic diseases such as allergic rhinitis, conjunctivitis, eczema and urticarial rashes (*Table 1*) and to have these conditions currently (*Table 2*). The prevalence of current exposure to cigarette smoke (in the household) was similar among cases and controls, although only about 11% were exposed (*Table 2*).

**Table 2.** Current risk factors and features of schoolchildren enrolled in an asthma case-control study between 2015–17 (N = 1,670).

| Current features | Asthma cases N = 555 | Non-asthma controls N = 1115 | Adj. OR (95% CI)[*] |
|---|---|---|---|
| Frequency of 'Trucks' passing on street near child's home | | | |
| Rarely (931) | 264 (47.6) | 667 (59.8) | 1 |
| Frequently (618) | 232 (41.8) | 386 (34.6) | 1.60 (1.27–2.01) |
| Almost all the time (121) | 59 (10.6) | 62 (5.6) | 2.28 (1.52–3.43)[†] |
| Cooking fuel most frequently used indoor | | | |
| No indoor cooking (390) | 120 (21.6) | 270 (24.2) | 1 |
| Charcoal stove (890) | 262 (47.2) | 628 (56.3) | 1.04 (0.79–1.37) |
| Gas/Electricity (390) | 173 (31.2) | 217 (19.5) | 1.58 (1.16–2.17) |
| Child's reported regular physical activity levels as recommended by WHO [m = 1] | | | |
| Yes (999) | 300 (54.0) | 699 (62.7) | 0.71 (0.56–0.90) |
| Current exposure to cigarette smoke by a household member[§] | | | |
| Yes (187) | 68 (12.2) | 119 (10.7) | 1.26 (0.90–1.76) |
| Use of de-worming medication in last 12 months [m = 1] | | | |
| None (570) | 151 (27.2) | 419 (37.6) | 1 |
| Once (724) | 231 (41.6) | 493 (44.3) | 1.37 (1.06–1.77) |
| ≥Twice (375) | 173 (31.2) | 202 (18.1) | 2.18 (1.62–2.93)[†] |
| Used albendazole in last 12 months [m = 18] | | | |
| Yes (892) | 352 (63.9) | 540 (49.0) | 1.76 (1.40–2.22) |
| Used praziquantel in last 12 months [m = 55] | | | |
| Yes (393) | 113 (21.0) | 280 (26.0) | 0.84 (0.63–1.11) |
| Infection with any helminths at enrolment[‡] [m = 127] | | | |
| Yes (206) | 53 (10.3) | 153 (14.9) | 0.75 (0.53–1.07) |
| Allergic rhinitis in last 12 months [m = 1] | | | |
| Yes (215) | 119 (21.4) | 96 (8.6) | 2.91 (2.13–3.97) |
| Allergic conjunctivitis in last 12 months | | | |
| Yes (177) | 86 (15.5) | 91 (8.2) | 2.10 (1.51–2.92) |
| Eczema in last 12 months | | | |
| Yes (60) | 41 (7.4) | 19 (1.7) | 4.52 (2.54–8.06) |
| Urticarial rash in last 12 months | | | |
| Yes (36) | 18 (3.2) | 18 (1.6) | 1.92 (0.96–3.84) |

Adj. OR = adjusted odds ratio; CI = confidence interval; N = number; m = missing; WHO = World Health Organisation. Number (%) in 2nd and 3rd column. [*]adjusted for child's age, sex, residence at birth and father's education level. [†]Test for trend p-value<0.0001. [‡]Helminths infections included mainly *Schistosoma mansoni*, *Trichuris trichuria*, Hookworm, and *Ascaris lumbricoides*; this variable was additionally adjusted for reported worm treatment in the last 12 months. [§]This included the adolescents' own smoking history.

## Atopy and asthma

Children with asthma were more likely to have allergic sensitisation: SPT positive to at least one of seven whole allergen extracts [2.40 (1.92–3.00)]; and elevated asIgE levels to any of three whole allergen extracts [2.45 (1.53–3.91)], and higher total IgE (*Table 3*). The most common allergens were dust mites and cockroach. Asthma cases were more likely to have elevated FENO levels [2.57 (2.01–3.29)] (*Table 3*).

## Assessment of different combinations of risk factors for asthma

We investigated the relative importance of area of residence at birth versus the first years of life, on the asthma risk, by looking at children who migrated between rural and urban areas during these two periods. Children born and raised in rural areas had the lowest risk (the reference group); children born and raised in urban areas had the highest risk [2.55 (1.80–3.61)], children born in the urban area who migrated and spent most of 0–5 years in rural areas still had an increased risk of asthma

**Table 3.** Atopy among schoolchildren enrolled in an asthma case-control between 2015–17 (N = 1,635).

| Atopy characteristics | Asthma cases N = 546 | Non-asthma controls N = 1089 | Adj. OR (95% CI)[‡] |
|---|---|---|---|
| Any positive skin prick test to any of 7 allergens | | | |
| Yes (653) | 302 (55.3) | 351 (32.2) | 2.40 (1.92–3.00) |
| *Blomia tropicalis* (dust mite) SPT | | | |
| Positive (478) | 241 (44.1) | 237 (21.8) | 2.51 (1.98–3.19) |
| *Dermatophagoides* mix (dust mite) SPT | | | |
| Positive (535) | 262 (48.0) | 273 (25.1) | 2.46 (1.95–3.10) |
| German cockroach SPT | | | |
| Positive (354) | 150 (27.5) | 204 (18.7) | 1.59 (1.23–2.06) |
| Peanut SPT [m = 4] | | | |
| Positive (57) | 27 (5.0) | 30 (2.8) | 1.73 (0.99–3.05) |
| Cat SPT | | | |
| Positive (22) | 22 (4.0) | 23 (2.1) | 1.94 (1.02–3.69) |
| Pollen mix (weeds) SPT [m = 2] | | | |
| Positive (43) | 22 (4.0) | 21 (1.9) | 2.14 (1.11–4.11) |
| Mould mix SPT | | | |
| Positive (19) | 8 (1.5) | 11 (1.0) | 1.40 (0.52–3.76) |
| Fractional exhaled nitric oxide (FENO) [m = 42] | | | |
| Elevated (≥35 ppb) (378) | 199 (36.7) | 179 (17.0) | 2.57 (2.01–3.29) |
| Any positive allergen-specific IgE[¥] (of 3 allergens, N = 392) | | | |
| Atopic level (249) | 143 (72.6) | 106 (54.4) | 2.45 (1.53–3.91) |
| *Dermatophagoides*-IgE | | | |
| Atopic level (186) | 116 (58.6) | 70 (35.9) | 2.32 (1.49–3.63) |
| Cockroach IgE | | | |
| Atopic level (198) | 110 (55.8) | 88 (45.1) | 1.69 (1.08–2.63) |
| Peanut IgE | | | |
| Atopic level (63) | 39 (19.7) | 24 (12.3) | 1.92 (1.05–3.50) |
| Total IgE: Mean (SD) kU/L | 930.25 (1246.60) | 640.88 (1146.61) | 1.00024 (1.00005–1.00043) |

N = number; Adj. OR = adjusted odds ratio; CI = confidence interval; SD = standard deviation; m = missing; ppb = parts per billion; number (%) shown in 2nd and 3rd column unless indicated otherwise.

[*]adjusted for child's age, sex, residence at birth and father's education level. Skin prick test using whole allergen extracts.

[¥]allergen-specific IgE levels were obtained using ImmunoCAP, on a random sample of 200 cases and 200 controls; standard cut-off for allergic sensitisation (≥0.35 allergen-specific kilo units per litre (kU$_A$/L)).

**Table 4.** Residence in early life as a risk factor for asthma among schoolchildren enrolled in a case-control study between 2015–17 (N = 1,670)

| Child's residence in early life | Asthma cases N = 555 | Non-asthma controls N = 1115 | Adj. OR (95% CI)* |
|---|---|---|---|
| Born and spent first 5 years in rural (265) | 50 (9.0) | 215 (19.3) | 1 |
| Born in rural, spent first 5 years in urban (87) | 24 (4.3) | 63 (5.6) | 1.54 (0.86–2.75) |
| Born in urban, spent first 5 years in rural (72) | 22 (4.0) | 50 (4.5) | 2.11 (1.14–3.91) |
| Born and spent first 5 years in urban (1,246) | 459 (82.7) | 787 (70.6) | 2.55 (1.80–3.61) |

N = number; Adj. OR = adjusted odds ratio; CI = confidence interval; number (%) shown in 2nd and 3rd column.

*Adjusted for age, sex, and father's education level.

[2.11 (1.14–3.91)], unlike children who were born in rural areas but migrated and spent most of 0–5 years in urban areas (*Table 4*).

We investigated the combined effects of the child's area of residence at birth and parental history of allergic disease. Even among children with no parental history of allergic disease, compared to being born in a rural area, being born in a town [2.15 (1.35–3.44)] or in the city [3.33 (1.61–6.90)] was associated with an increased risk of asthma, *Table 5*. Compared to the same reference group, children with a parental history of allergic disease had a higher risk of asthma that increased steadily from among children born in rural areas [2.78 (1.58–4.89)], in a town [5.16 (3.25–8.17)] and in the city [5.54 (2.99–10.26)] (*Table 5*).

We also investigated the combined effects of area of residence at birth and children's atopic status (positive SPT to any of seven allergens). Taking non-atopic children born in rural areas as the reference group, we found that non-atopic children born in town had a modest increase in asthma risk [1.72 (1.16–2.55)], which increased further among city-born children [2.63 (1.41–4.91)], *Table 6*. However, atopic children had a modestly increased risk of asthma even if they were born in the rural area [1.77 (1.02–3.05)], which increased substantially among atopic children born in town [4.61 (3.07–6.90)] or in the city [4.67 (2.56–8.52)], *Table 6*.

We investigated the combined effects of parental education and urbanicity on asthma risk, by looking at the father's education level in combination with the child's area of residence at birth. We found that for each level of father's education, there was an increasing risk gradient from rural-town-city; for each area of residence, increasing father's education level was associated with increased asthma risk; the highest risk of asthma was among children born in the city whose fathers had a tertiary education [6.95 (3.52–13.71)] (*Table 7*). The same pattern in *Table 7* was seen for mother's education and for child's area of residence in the first five years of life.

For the combined effect of atopy (SPT positive) and parental history of allergic disease, we found that asthma risk was more than five times among children with both atopy and a parental history of allergic disease [5.74 (4.07–8.10)] (*Supplementary file 1b*). However, children who had both parents

**Table 5.** Combined effects of residence at birth and parental history of allergic disease as risk factors for asthma among schoolchildren in a case-control study from 2015 to 17 (N = 1,532).

| Child's residence at birth | Parental history of allergy | Asthma cases N = 526 | Non-asthmatic controls N = 1006 | Adj. OR (95% CI)* |
|---|---|---|---|---|
| Rural | – | 27 (5.1) | 158 (15.7) | 1 |
| Town | – | 137 (26.1) | 383 (38.1) | 2.15 (1.35–3.44) |
| City | – | 20 (3.8) | 31 (3.1) | 3.33 (1.61–6.90) |
| Rural | + | 42 (8.0) | 85 (8.4) | 2.78 (1.58–4.89) |
| Town | + | 253 (48.1) | 306 (30.4) | 5.16 (3.25–8.17) |
| City | + | 47 (8.9) | 43 (4.3) | 5.54 (2.99–10.26) |

Adj. OR = adjusted odds ratio; CI = confidence interval; N = number; '−' refers to no history of parental allergy, '+' refers to positive history of parental allergy. number (%) in 3rd and 4th column. *Adjusted for child's age, sex, and father's education level. Parental history of allergic disease included a history of asthma, eczema, allergic rhinitis, allergic conjunctivitis and any other allergies.

**Table 6.** Combined effects of residence at birth and atopy as asthma risk factors among schoolchildren in a case-control study between 2015–17 (N = 1,635).

| Child's residence at birth | Atopy (SPT to any of 7 allergens) | Asthma cases N = 546 | Non-asthmatic controls N = 1089 | Adj. OR (95% CI)* |
|---|---|---|---|---|
| Rural | - | 42 (7.7) | 191 (17.5) | 1 |
| Town | - | 177 (32.4) | 504 (46.3) | 1.72 (1.16–2.55) |
| City | - | 25 (4.6) | 43 (4.0) | 2.63 (1.41–4.91) |
| Rural | + | 31 (5.7) | 81 (7.4) | 1.77 (1.02–3.05) |
| Town | + | 228 (41.7) | 236 (21.7) | 4.61 (3.07–6.90) |
| City | + | 43 (7.9) | 34 (3.1) | 4.67 (2.56–8.52) |

Adj. OR = adjusted odds ratio; CI = confidence interval; N = number; SPT = skin prick test; '-' refers to SPT negative, '+' refers to SPT positive to any of seven crude extracts of *Blomia tropicalis*, Dermatophagoides mix, cockroach, peanut, cat, weeds pollen mix, and mould mix. number (%) in 3rd and 4th column. *Adjusted for child's age, sex and father's education level.

with a history of allergic disease had an effect size similar to children who had only one parent with a history of allergic disease [2.49 (1.74–3.58)] (*Supplementary file 1c*).

## Discussion

We found a step-wise increase in asthma risk according to the child's area of residence at the time of birth and in their first five years of life. Children born and raised in rural areas had the lowest risk, children born and raised in small towns had a 2-fold increase in risk, while children born and raised in the city had a 3-fold increase in asthma risk. This is the first study in Africa to show such a strong gradient in asthma risk by place of birth. Previous studies have been mostly cross-sectional and conducted in either rural or urban settings, and therefore focusing on current residence; these have shown that the prevalence of asthma is lower among rural residents compared to urban residents (*Lawson et al., 2017*; *Morgan et al., 2018*; *Botha et al., 2019*).

Our findings confirm that the area where a child is born (usually the same as the area where the mother was resident during pregnancy) is important for asthma risk. Indeed, we found that when children moved to other environments, they carried with them the asthma risk related to their area of residence in pregnancy: children born in urban areas who were subsequently raised in rural areas still had a 2-fold increase in asthma risk, similar to their counterparts born and raised in urban areas. These findings are comparable to observations from Europe showing that children born on a farm (usually in rural areas) have a lower risk of asthma in later life, even when they subsequently moved to urban areas (*Leynaert et al., 2001*), and that children who migrated to Europe after age five had

**Table 7.** Combined effects of father's education and residence at birth as asthma risk factors among schoolchildren in a case-control study from 2015 to 17 (N = 1,670).

| Father's education level | Child's residence at birth | Asthma cases N = 555 | Non-asthma controls N = 1115 | Adj. OR (95% CI)* |
|---|---|---|---|---|
| Primary | Rural | 20 (3.6) | 115 (10.3) | 1 |
| Primary | Town | 67 (12.1) | 206 (18.5) | 2.13 (1.22–3.74) |
| Primary | City | 12 (2.2) | 17 (1.5) | 4.98 (2.02–12.31) |
| Secondary | Rural | 26 (4.7) | 109 (9.8) | 1.39 (0.72–2.66) |
| Secondary | Town | 156 (28.1) | 268 (24.0) | 3.97 (2.34–6.76) |
| Secondary | City | 11 (1.9) | 24 (2.2) | 2.80 (1.16–6.76) |
| Tertiary | Rural | 28 (5.0) | 54 (4.8) | 3.22 (1.64–6.33) |
| Tertiary | Town | 189 (34.1) | 283 (25.4) | 4.96 (2.89–8.53) |
| Tertiary | City | 46 (8.3) | 39 (3.5) | 6.95 (3.52–13.71) |

N = number; CI = confidence interval; Adj. OR = adjusted odds ratio; 3rd and 4th column contain n (%); *Adjusted for child's age and sex. Similar pattern was observed for mother's education level and child's residence at birth.

a lower prevalence of asthma, similar to their country of origin, than children who were either born in Europe or migrated before the age of five (*Migliore et al., 2007*; *Kuehni et al., 2007*). However, there are also studies that show increased risk of wheezing and allergy among children following migration (*Rodriguez et al., 2017*; *Stein et al., 2016*). Unlike studies from Europe and North America which have reported a lower risk of asthma for children born or raised on farms (*Genuneit, 2012*; *Timm et al., 2015*), our study found no association between asthma risk and exposure to farm animals either during pregnancy or in early life. We hypothesise that this may be due to ubiquitous farm animal exposure due to subsistence farming, a widespread practice in Uganda, even in towns.

Our observation that children with asthma were more likely to have parents with tertiary education and to use gas or electricity for indoor cooking (as opposed to charcoal stoves) has been made by an earlier study in Uganda (*Nantanda et al., 2013*). Similarly, our finding that children with asthma reported the highest frequency of 'trucks passing on the street near their home' has been reported elsewhere (*Sharma and Banga, 2007*; *Shirinde et al., 2014*; *Venn et al., 2005*). We suggest that these factors are proxy measures of a higher socio-economic status of asthma cases and of urbanicity, consistent with findings from other LMICs that have found a higher prevalence of asthma among children (and adults) in urban than rural areas (*Morgan et al., 2018*; *Robinson et al., 2011*; *Gaviola et al., 2016*). However, our findings contradict those from HICs in which asthma is associated with lower parental education (*Lewis et al., 2017*) and socio-economic status (*Akinbami et al., 2011*). This suggests that there may be similarities in lifestyle and environmental factors between the highly educated and high socio-economic status families in LMICs with the low educated and low socio-economic status families in HICs, which increase asthma risk, and therefore require further investigation.

Although children with asthma were more likely to be sensitised to allergens than controls, the pattern of allergic sensitisation was similar among cases and controls; majorly sensitised to house-dust mites and cockroach, and least sensitised to peanut, cat, pollen and mould. This SPT response pattern was similar to other studies from Africa (*Nyembue et al., 2012*; *Mbatchou Ngahane et al., 2016*), but different from Europe where the main allergens are dust mite, cat and pollen (*Bousquet et al., 2007*).

Although maternal smoking is a known risk factor for childhood asthma (*Silvestri et al., 2015*), our study found no association between maternal smoking and asthma. We attribute this to the low prevalence of smoking in this population, by the mothers during pregnancy (2.5%) and by any household members currently (11%). The lack of association between asthma and indoor cooking with biomass fuels in this study is consistent with other studies from Africa (*Thacher et al., 2013*; *Oluwole et al., 2017a*). Indeed, this is consistent with the general pattern of lower risk of asthma in rural areas (where biomass fuel use is highest) than urban areas.

We found an inverse association between asthma and current infection with any helminths, particularly *T. trichuria*. However, the association between asthma and current infections in Africa has been inconsistent across studies (*Mpairwe and Amoah, 2019*). What was novel in this study is that we also collected data on history of using de-worming medication in the last 12 months. We found that children with asthma were more likely to have used de-worming medication more than twice in the last 12 months compared to controls, and this was de-worming with albendazole (for geo-helminths) not with praziquantel (for schistosomiasis). This implies that the inverse association we noted between asthma and helminths may be partly explained by increased de-worming among children with asthma. The current de-worming schedule in this age-group in Uganda includes mass-deworming in schools, bi-annually for albendazole and once a year for praziquantel. We did not establish whether the additional de-worming was self-medicated or prescribed by medical workers.

We found that children with asthma were less likely to have a BCG scar, but there was no association with tuberculin skin test at enrolment. In Uganda, BCG vaccination is routinely given at birth. A lack of BCG scar does not mean the vaccine was not administered, but may indicate differences in immune responses among children who will eventually develop asthma. Alternatively, BCG vaccination may be protective against asthma, as suggested by previous studies (*El-Zein et al., 2010*).

The other important risk factors for asthma, that have been previously described, included parental history of allergic disease (*Lim et al., 2010*), a child's atopy status and concomitant other allergic disease (*Bao et al., 2017*). The strength of this study was to demonstrate that having a combination of any two of these known risk factors for asthma had an additive effect on asthma risk, and that this risk also increased in relation to area of residence in early life, with an increasing rural-town-city

gradient. This gradient and independent effects of parental education provide strong evidence for the role of environmental and lifestyle factors associated with urbanisation that are responsible for the increasing asthma risk in urban areas. More investigations are required to identify the specific factors, in order to design interventions to modify them so as to prevent the establishment of asthma risk in early life.

This study had limitations inherent to all case-control studies, such as potential recall bias, selection bias and confounding. We minimised recall bias by focusing on major early life events that a parent was likely to remember. It was re-assuring to note a strong correlation between the recalled events and objective measures such as SPT. We minimised the selection bias for controls by randomly selecting controls from the same class register where the cases were obtained, and by randomly selecting the 400 participants for the asIgE assay. We minimised confounding by adjusting for measured confounders in all our analyses, but cannot rule out the possible role of unmeasured confounders. Finally, a large number of potential participants were not included in this study, mostly because they were in the boarding school section and were unable to contact their parents/guardians in time to attend the parents' meeting (to provide consent). We do not have information on whether these parents were more likely to reside in rural areas or the city, where the asthma risk is either lower or higher than the town where this study was conducted, respectively. Our results may or may not be generalisable to similar urban areas in Uganda and Sub-sahara Africa.

Our findings support the concept that environmental factors during early life may influence the risk of development of NCDs in later life (*Fleming et al., 2018*). However, our study does not involve information sufficiently detailed to identify whether exposures in utero or in early life are most relevant. Further research in this setting is required to investigate these hypotheses, and this would have important implications for the life course approach in the prevention of asthma globally.

## Conclusion

The risk of asthma among schoolchildren in urban Uganda is strongly predicted by their area of residence in early life, particularly at birth, with the highest risk among children whose early life is spent in small towns and in the city. This risk increases further in the presence of other asthma risk factors such as parental history of allergic disease, children's own atopy, higher measures of socio-economic status and urbanicity. Given the current rapid urbanisation in Africa and other LMICs, the prevalence of asthma is likely to increase further. This study provides the basis for future studies investigating environmental and lifestyle factors that increase asthma risk in the urban areas of LMICs.

# Materials and methods

**Key resources table**

| Reagent type (species) or resource | Designation | Source or reference | Identifiers | Additional information |
|---|---|---|---|---|
| Commercial assay or kit | ImmunoCAPspecific IgE test | Thermo Fisher Scientific, Uppsala, Sweden | | http://www.phadia.com/en/Products/ Allergy-testing-products/ImmunoCAP-Assays/sIgE/ |

## Study design

We conducted an un-matched case-control study among schoolchildren, and report following STROBE guidelines (*von Elm et al., 2014*).

## Study population, sampling and consent

The study base was determined a priori as schoolchildren, 5–17 years old, in primary and secondary schools in Entebbe Municipality and Katabi zone in Wakiso District, Central Uganda. This was a predominantly urban (town) setting. All schools in the study area were invited and 96% participated. Study enrolment was between May 2015 and July 2017. We estimated that a sample size of 2112 would have 80% power to detect odds ratio (OR) <0.5 or>1.5 for exposures with prevalence 10% among controls.

At each school, we pre-screened by registering all children with any breathing problems. We concurrently randomly selected from the class register two children without any breathing problems, using a random number generator programme in STATA (StataCorp, Texas, USA). The children

delivered invitation letters to their parents; parents/guardians with telephones were invited to attend a meeting during which those interested provided written informed consent, and children aged $\geq 8$ years provided written informed assent.

## Definition of cases and controls

Following consent and assent, we screened all participants with the International Study of Asthma and Allergies in Childhood (ISAAC) questionnaire (*Asher et al., 1995*). Children with a history of wheezing in the last 12 months underwent a detailed medical history and examination by study clinicians, to diagnose asthma. Asthma was defined as a history of recurrent symptoms of wheezing, cough (mostly dry, worse at night and morning) and/or difficulty in breathing experienced in the last 12 months, with or without forced expiratory volume in the first second ($FEV_1$) $\leq 80\%$ expected for age, sex and height for African children. Additional medical history included a prior physician diagnosis and good response to asthma medication. If the diagnosis was not straightforward, two clinicians reviewed the participant and if they disagreed, that participant was excluded in order to ensure proper case ascertainment. Children with a history of wheeze or any asthma symptoms but not in the last 12 months were excluded. Controls were defined as having no history of wheeze or any other asthma symptoms. There were no other exclusion criteria.

## Ethical approval

The study was approved by the Uganda Virus Research Institute Research and Ethics Committee, and the Uganda National Council for Science and Technology.

## Clinical assessments

We collected data about asthma risk factors and allergic conditions identified in literature using interviewer-led questionnaires to parents and adolescents, including the ISAAC questionnaire (*Asher et al., 1995*). When a parent was not available in person and the participant answering the questionnaire did not know that parent's history of allergy, we telephoned the parent for the relevant information. We looked for the presence of a Bacillus Calmette–Guérin (BCG) scar since BCG vaccination has been inversely associated with asthma previously (*El-Zein et al., 2010*). We tested for fractional exhaled nitric oxide (FENO), using a hand-held device (NoBreath from Bedfonf Scientific, Maidstone, United Kingdom), and used the manufacturer's cut-off for children of $\geq 35$ parts per billion. FENO is considered a biomarker of allergic airway inflammation (*Hoyte et al., 2018*). We conducted lung function tests for asthma cases using a hand-held spirometer (Micro 1 Diagnostic Spirometer, CareFusion, Chatham Marine, United Kingdom).

Skin prick tests (SPT) and allergen-specific IgE (asIgE) are important measures of allergic sensitisation (*Heinzerling et al., 2013*). We conducted (SPT) following standard procedures (*Heinzerling et al., 2013*; *Mpairwe et al., 2008*), and crude extracts of seven allergens (*Dermatophagoides* mix of *D. farinae* and *D. pteronyssminus*, *Blomia tropicalis*, *Blattella germanica*, peanut, cat, pollen mix of weeds, mould mix of *Aspergillus* species; ALK Abello, Hoersholm, Denmark). A positive response was a wheal diameter $\geq 3$ mm measured after 15 min, with a negative saline control and positive histamine. We collected blood samples which we processed to obtain plasma that we stored at $-80°C$. At the end of the study, we randomly selected 200 plasma aliquots from all asthma cases and 200 from all controls for measurement of asIgE to whole allergen extracts (*D. pteronyssinus*, *B. germanica* and peanut), using ImmunoCAP (Phadia, Uppsala, Sweden). The standard cut-off for allergic sensitisation of $\geq 0.35$ allergen-specific kilo units per litre ($kU_A/L$) was used (Key resources table).

Because previous studies have reported an association between asthma and helminths (*Mpairwe and Amoah, 2019*), we collected three fresh stool samples from each participant and tested for intestinal helminths using the Kato Katz method (*Katz et al., 1972*). The findings on the association between tuberculin skin test (TST) and allergy/asthma have been inconsistent (*Obihara et al., 2006*; *Eifan et al., 2009*). We investigated this association by performing the TST using standard procedures, as we have previously described (*Nkurunungi et al., 2012*).

## Data management and statistical analysis

Data were double-entered using OpenClinica open source software version 3.1.4 (OpenClinica LLC and collaborators, Waltham, MA, USA). Data were analysed in STATA.

For continuous variables with clinically relevant cut-off points such as for SPT, asIgE, FENO and TST, we used the dichotomous variables in the analysis. Total IgE was analysed as a continuous variable. The variable for maternal or paternal history of 'allergic disease' combines the history of asthma, eczema, allergic rhinitis, allergic conjunctivitis and any other allergies (*Greenland et al., 2016*). We conducted a complete case analysis, and did not impute missing values.

The key variables adjusted for (see below) were age, sex, area of residence at time of birth and father's education. Each of these variables (except sex) had small numbers of missing values (the largest number was 29 for father's education). We therefore created a 'complete case' data set, comprising the 555 cases and 1115 controls which had no missing values for these variables. This 'complete case' data set was used for all analyses.

The odds ratio was the main outcome measure (as is appropriate for a case-control study), and we also estimated the 95% CIs. We identified age and sex as a priori confounders, and all analyses were adjusted for these variables. We identified area of residence at time of birth and father's education as potential confounders; we did not also adjust for area where the child spent most of the first five years or mother's education, since these were strongly associated with the above two factors, and therefore would have introduced collinearity. Previous studies in this setting have found that father's and mother's education are significantly associated with socioeconomic status (*Aarø et al., 2009*).

We did not identify any factors which were likely to be on the causal pathways (and therefore were not potential confounders and should not be adjusted for). We conducted initial logistic regression analyses for each exposure of interest, adjusted for age and sex, with random effects to allow for any clustering by school. We then ran the adjusted random effects logistic regression model (i.e. adjusted for the above factors in addition to age and sex) for each exposure of interest, and checked for collinearity by comparing the standard error of the exposure coefficient in the adjusted model and in the basic model (*Greenland et al., 2016*; *Greenland and Pearce, 2015*). We did not find any problems of collinearity, so we reported the findings for the adjusted model for each exposure.

## Acknowledgements

We thank the study participants, their parents and guardians for their enthusiastic participation. We thank the teachers and school administrations for providing us with a conducive environment in which to conduct this study. We acknowledge the support we received from the Entebbe Municipality and Wakiso District Education Officials. Many thanks to Kisubi Hospital staff, particularly Dr. Robert Asaba and Dr. Rogers Sendijja, for their support during the results dissemination meetings we held in their premises. Many thanks to Ronald van Ree from Leiden University Medical Centre (The Netherlands) for working with GN on the asIgE ImmunoCAP assays.

## Additional information

### Funding

| Funder | Grant reference number | Author |
| --- | --- | --- |
| Wellcome | Training fellowship 102512 | Harriet Mpairwe |
| Wellcome | Senior fellowship 095778 | Alison M Elliott |
| European Research Council | Project grant 668954 | Neil Pearce |

The funders had no role in study design, data collection and interpretation, or the decision to submit the work for publication.

### Author contributions

Harriet Mpairwe, Conceptualization, Formal analysis, Funding acquisition, Investigation, Visualization, Writing—original draft, Project administration, Writing—review and editing; Milly Namutebi,

Investigation, Project administration, Writing—review and editing; Gyaviira Nkurunungi, Pius Tumwesige, Irene Nambuya, Mike Mukasa, Caroline Onen, Marble Nnaluwooza, Barbara Apule, Tonny Katongole, Gloria Oduru, Investigation, Writing—review and editing; Joseph Kahwa, Data curation, Writing—review and editing; Emily L Webb, Lawrence Lubyayi, Formal analysis, Writing—review and editing; Neil Pearce, Supervision, Funding acquisition, Writing—review and editing; Alison M Elliott, Conceptualization, Supervision, Funding acquisition, Writing—review and editing

### Author ORCIDs
Harriet Mpairwe (iD) https://orcid.org/0000-0003-1199-4859
Gyaviira Nkurunungi (iD) http://orcid.org/0000-0003-4062-9105

### Ethics

Human subjects: Parents or guardians of the children provided written informed consent, and children eight years or older provided written informed assent. This consent was to participate in the study, and to publish anonymous results. The study was approved by the Uganda Virus Research Institute Research and Ethics Committee, and the Uganda National Council for Science and Technology [reference number HS 1707]. The two bodies follow Good Clinical Practice guidelines.

### Decision letter and Author response
Decision letter https://doi.org/10.7554/eLife.49496.sa1
Author response https://doi.org/10.7554/eLife.49496.sa2

## Additional files

### Supplementary files
• Supplementary file 1. Tables of results for risk factors for asthma among schoolchildren involved in an asthma case-control study in Uganda, between 2015–17. *Supplementary file 1a* The association between infection with different species of helminths and asthma among 1543 schoolchildren. Three fresh stool samples per child were examined for helminths using the Kato Katz method. We used multiple logistic regression method, adjusted for child's age, sex, residence at birth, father's education level and reported worm treatment in the last 12 months. [a]Other helminth infections included *Hymenolepis nana* and *Enterobius vermicularis.* N = number; Adj. OR = adjusted odds ratio; CI = confidence interval; number (%) in 2nd and 3rd column. *Supplementary file 1b* The association between individual and combined effects of child's atopy and parental history of allergic disease, and asthma risk among 1501 schoolchildren. Skin prick test (SPT) performed using standard procedures and seven crude extracts of *Blomia tropicalis*, Dermatophagoides mix, cockroach, peanut, cat, weeds pollen mix, and mould mix. Parental history of allergic disease included a history of asthma, eczema, allergic rhinitis, allergic conjunctivitis and any other allergies. We used multiple logistic regression analysis, and adjusted for child's age, sex, residence at birth and father's education level. N = number; Adj. OR = adjusted odds ratio; CI = confidence interval; '-' refers to none, '+' refers to present. Number (%) in 3rd and 4th column. *Supplementary file 1c* The individual and combined effects of mother's and father's history of allergic disease, and asthma risk among 1498 schoolchildren. Parental history of allergic disease included a history of asthma, eczema, allergic rhinitis, allergic conjunctivitis and any other allergies. We conducted multiple logistic regression analysis, and adjusted for child's age, sex, residence at birth and father's education level. N = number; Adj. OR = adjusted odds ratio; CI = confidence interval; number (%) shown in 2nd and 3rd column.

• Reporting standard 1. STROBE Statement: Risk factors for asthma among schoolchildren who participated in a case-control study in urban Uganda.

• Transparent reporting form

### Data availability
Data is available on request via https://datacompass.lshtm.ac.uk/1369/. To gain access to the data please complete the application process via the website. Requests will be reviewed and assessed by the corresponding author, in consultation with the LSHTM's Research Data Manager and relevant

LSHTM staff members responsible for research governance and data protection. Applications will be evaluated on the basis of their compatibility with the study's research objectives and the ability to provide de-identified data sufficient to meet the intended purpose, without breaching participant confidentiality or the study's ethical and legal commitments. Successful applicants will be asked to sign a Data Transfer Agreement prior to being provided with the data.

The following dataset was generated:

| Author(s) | Year | Dataset title | Dataset URL | Database and Identifier |
|---|---|---|---|---|
| Webb E, Mpairwe H | 2019 | SONA project - Asthma risk factors data | https://doi.org/10.17037/DATA.00001369 | London School of Hygiene & Tropical Medicine (LSHTM) Data Compass, 10.17037/DATA.00001369 |

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
