## [Decision Letter]

**Acceptance summary:**

There are only a few studies on asthma from African countries and this is particularly true of Uganda. The authors conducted a study to identify the main risk factors for asthma among schoolchildren living in urban Uganda. The authors show that the places of birth and residence in the early years of life contribute to the risk of asthma in later years. The findings raise important questions on early environmental and lifestyle determinants of asthma, specifically among children living in poverty and in low- and middle-income countries. This work is therefore of interest to the researchers working on asthma and early determinants of health.

**Decision letter after peer review:**

Thank you for submitting your article "Risk factors for asthma among schoolchildren who participated in a case-control study in urban Uganda" for consideration by *eLife*. Your article has been reviewed by three peer reviewers, one of whom is a member of our Board of Reviewing Editors, and the evaluation has been overseen by Eduardo Franco as the Senior Editor. The following individuals involved in review of your submission have agreed to reveal their identity: Akhil Soman (Reviewer #3).

The reviewers have discussed the reviews with one another and the Reviewing Editor has drafted this decision to help you prepare a revised submission.

Summary:

There are only a few studies on asthma from African countries and this is particularly true of Uganda. Therefore, this paper will add to the sparse literature. Using a case control design, the authors have shown that the places of birth and residence in the early years of life contribute to the risk of asthma in later years. The paper is generally well written. However, there are a few methodological and analytic issues that limit the contribution of the paper to the literature. The following review lists several items in need of revision.

Essential revisions:

Introduction:

1) With respect to scarcity of evidence on asthma risk factors from Africa, the authors attempt to address an important gap in the literature. However, the current Introduction does not provide a strong rationale for the analysis undertaken. It would be worthwhile to more comprehensively summarise the relevant literature by elaborating on the African context of this study through greater description of existing knowledge and explicit comparisons with studies from other African countries. The paper would also benefit from comparisons with studies from other low and middle income countries.

Materials and methods:

1) It would help to logically sequence the Materials and methods section. The type of case-control study, the secondary study base, time period of study and case and control size need to be highlighted at the start of the Materials and methods section. A helpful participant flow diagram is presented in Figure 1. However, in the current text, sample size is introduced in results and sampling of controls comes in data management and statistical analysis.

2) Two sources of interviews were used: in person and telephone interviews; does this mean that only children whose parents had a telephone were included in the study? Also, using two methods of interview can lead to differential responses. What was the distribution of this differential interview pattern? Can the authors comment on the potential of information bias arising from this pattern of interview? There are also some concerns related to pre-screening procedures: how confident are the authors that the children reported breathing problems accurately? The age range (5-17) is wide; this increases the possibility of obtaining erroneous information.

3) The several examinations mentioned under clinical assessment may be important for the topic addressed. Yet the rationale or relevance of conducting these examinations (e.g. BCG scar, exhaled nitric oxide, skin prick tests, tests for intestinal helminths, TST) is not clearly stated. In addition, a definition and diagnostic criteria for asthma should be provided. Another point related to data collection was not mentioned: was there information on smoking habits of the children/adolescents? This is especially important for secondary school students.

4) Only 28.7% of the original sample agreed to participate in the study; could you please elaborate on the potential impact of this low response rate on the results of study? Also, what was the difference in the response rate between cases and controls?

Analytic strategy:

The statistical analysis section should more coherently present confounders and mediators identified among the measured variables, statistical models used, and measures of strength and precision used to describe results. Below are a few specific comments that should either be clarified or considered through the conduct of further analysis:

1) One of the major concerns is that the sample in the study is drawn from 50+ schools and thus prone to clustering effects. There is no indication in the data analysis of whether this issue was considered. We strongly recommend that the authors address and justify what they have done. If they cannot, please redo the analyses.

2) The authors state that they have assessed combined effects of risk factors on additive and multiplicative scales. They have also interpreted the results given in tables under the combined effect Results section as additive or multiplicative. However, it can be argued that the authors do not present sufficient results to make such interpretations. What were the measures and associated confidence intervals based on and which combined effect on additive or multiplicative scales were assessed? P-values are not enough to estimate strength and precision of interaction results. If the authors have not used any such measures, it is recommended to just present stratum specific effects of these exposures and refrain from making any claims about interaction. Please highlight these in the limitations.

3) While we appreciate that the causal pathway factors were not adjusted in the models, there is no explanation of what those factors were. Was there any technique used (e.g., causal graphs) to identify them?

4) The paper would benefit from an explanation for the choice of socioeconomic indicator used in the final models (Why father's education? Were the continuous variables used as linear functions in the models? Was the potential for non-linearity tested? Several exposures were tested against a single outcome. Did the authors correct for multiple comparisons? If not, why?

5) The 95% CIs were estimated in the analysis. However, this is not stated in the analysis section. Also, please clarify the relevance of p-values (again not stated in the Materials and methods section) in the results tables. There is a strong argument in the literature that estimates and CIs, not p-values, convey the strength and precision of results (please refer to Nature comment: "Scientists rise up against statistical significance", 20 March 2019)

6) Table 1 shows missingness (ranging from N=0 to N=420) related to several variables. How did the authors deal with missing values? Were imputation and sensitivity analysis considered? Any comments related to potential bias?

[Editors' note: further revisions were requested prior to acceptance, as described below.]

Thank you for resubmitting your work entitled "Risk factors for asthma among schoolchildren who participated in a case-control study in urban Uganda" for further consideration by *eLife*. Your revised article has been evaluated by Eduardo Franco (Senior Editor) and Belinda Nicolau (Reviewing Editor).

The manuscript has been improved but there are some remaining issues that need to be addressed before acceptance, as outlined below:

1) In subsection “Current features of asthma cases versus controls”, the authors say that there was weak evidence for an inverse association between asthma and infection with any helminth's species. Based on the odds ratio and confidence interval presented, we suggest the authors to further elaborate this conclusion.

2) Please check the order of sections in the manuscript, table numbers, titles etc.

3) Please check for duplicates in reference, e.g., 53 and 56.

---

## [Author Response]

Essential revisions:Introduction:1) With respect to scarcity of evidence on asthma risk factors from Africa, the authors attempt to address an important gap in the literature. However, the current Introduction does not provide a strong rationale for the analysis undertaken. It would be worthwhile to more comprehensively summarise the relevant literature by elaborating on the African context of this study through greater description of existing knowledge and explicit comparisons with studies from other African countries. The paper would also benefit from comparisons with studies from other low and middle income countries.

We thank the reviewer for pointing out ways we could improve the Introduction section. We have added more relevant literature from Africa and other LMICs, mindful not to make the introduction too long.

“There is little previously published data on asthma risk factors from Africa. Unlike in HICs, higher parental education and socioeconomic status has been associated with asthma among children in Africa (Addo-Yobo et al., 2007; Nantanda et al., 2013;, Wolff et al., 2012).”

Materials and methods:1) It would help to logically sequence the Materials and methods section. The type of case-control study, the secondary study base, time period of study and case and control size need to be highlighted at the start of the Materials and methods section. A helpful participant flow diagram is presented in Figure 1. However, in the current text, sample size is introduced in results and sampling of controls comes in data management and statistical analysis.

In accordance to the reviewer’s suggestions, we have made changes to the start of the Materials and methods section, which now reads as shown below.

Study design

We conducted an un-matched case-control study among schoolchildren, and report following STROBE guidelines (von Elm et al., 2014).

Study population, sampling and consent

The study base was determined a priori as schoolchildren, 5-17 years old, in primary and secondary schools in Entebbe Municipality and Katabi zone in Wakiso District, Central Uganda. This was a predominantly urban (town) setting. All schools in the study area were invited and 96% participated. Study enrolment was between May 2015 and July 2017. We estimated that a sample size of 2,112 would have 80% power to detect odds ratio (OR) <0.5 or >1.5 for exposures with prevalence 10% among controls.

At each school, we pre-screened by registering all children with any breathing problems. We concurrently randomly selected from the class register two children without any breathing problems, using a random number generator programme in STATA (StataCorp, Texas, USA).”

“Following consent and assent, we screened all participants with the International Study of Asthma and Allergies in Childhood (ISAAC) questionnaire (Asher et al., 1995).”

2) Two sources of interviews were used: in person and telephone interviews; does this mean that only children whose parents had a telephone were included in the study? Also, using two methods of interview can lead to differential responses. What was the distribution of this differential interview pattern? Can the authors comment on the potential of information bias arising from this pattern of interview?

We agree that the issue of telephone interviews requires more clarification. We only telephoned a small proportion of children’s mothers (6.6%) and fathers (3.4%) to specifically answer the question regarding their history of allergy. This small proportion of telephone interviews is unlikely to have introduced any information bias to the study. We have clarified this point more in the Materials and methods section.

“When a parent was not available in person and the participant answering the questionnaire did not know that parent’s history of allergy, we telephoned the parent for the relevant information.”

Table 1 legend: “For parental history of allergy, 6.6% mothers and 3.4% fathers responded by telephone.”

There are also some concerns related to pre-screening procedures: how confident are the authors that the children reported breathing problems accurately?

It is true that the children may not have reported breathing problems accurately at the pre-screening stage. However, the purpose of the pre-screening stage was to identify all children with any breathing problems in our primary study base. Then, during the screening and enrolment stages of the study, proper case ascertainment was made by the clinical study team. We are confident of the accuracy of the final diagnosis of the asthma cases.

“If the diagnosis was not straightforward, two clinicians reviewed that participant and if they disagreed, the participant was excluded in order to ensure proper case ascertainment.”

The age range (5-17) is wide; this increases the possibility of obtaining erroneous information.

We agree with the reviewer that age 5-17 is a wide range. In order to minimise the possibility of obtaining erroneous information, we used interviewer-led questionnaires for both adolescents and parents (who answered on behalf of the younger children).

3) The several examinations mentioned under clinical assessment may be important for the topic addressed. Yet the rationale or relevance of conducting these examinations (e.g. BCG scar, exhaled nitric oxide, skin prick tests, tests for intestinal helminths, TST) is not clearly stated.

We agree with the reviewer and have now rewritten the clinical assessments section to include the rationale, with relevant references, for the various tests that we conducted.

“We looked for the presence of a Bacillus Calmette–Guérin (BCG) scar since BCG vaccination has been inversely associated with asthma previously (El-Zein et al., 2010).”

“ENO is considered a biomarker of allergic airway inflammation (Hoyte, Gross and Katial, 2018).”

“Skin prick tests (SPT) and allergen-specific IgE (asIgE) are important measures of allergic sensitisation (Heinzerling et al., 2013).”

“Because previous studies have reported an association between asthma and helminths (Mpairwe and Amoah, 2019), we collected three fresh stool samples from each participant and tested for intestinal helminths using the Kato Katz method (Katz, Chaves and Pellegrino, 1972). The findings on the association between tuberculin skin test (TST) and allergy/asthma have been inconsistent (Obihara et al., 2006; Eifan et al., 2009). We investigated this association by performing the TST using standard procedures, as we have previously described (Nkurunungi et al., 2012).”

In addition, a definition and diagnostic criteria for asthma should be provided.

Thank you for pointing out this important omission. We have now clarified the definition and diagnostic criteria for asthma.

“Following consent and assent, we screened all participants with the International Study of Asthma and Allergies in Childhood (ISAAC) questionnaire (Asher et al., 1995). Children with a history of wheezing in the last 12 months underwent a detailed medical history and examination by study clinicians, to diagnose asthma. Asthma was defined as a history of recurrent symptoms of wheezing, cough (mostly dry, worse at night and morning) and/or difficulty in breathing experienced in the last 12 months, with or without forced expiratory volume in the first second (FEV_1_) <80% expected for age, sex and height for African children. Additional medical history included a prior physician diagnosis and good response to asthma medication. If the diagnosis was not straightforward, two clinicians reviewed that participant and if they disagreed, the participant was excluded in order to ensure proper case ascertainment.”

Another point related to data collection was not mentioned: was there information on smoking habits of the children/adolescents? This is especially important for secondary school students.

It is true that smoking is important in asthma. We collected data on current smoking by any household member (including the adolescents themselves) and the overall prevalence was only 11% and did not differ between asthma cases and controls (Table 2, rows 10-11).

Generally, details of the items from the questionnaires are presented in the results tables. We did this to keep the Materials and methods sections short and simple. But we are willing to include details of questionnaire items in the Materials and method’s section as well, if that is preferred.

“This included the adolescents’ own smoking history.” added to Table 2 legend.

4) Only 28.7% of the original sample agreed to participate in the study; could you please elaborate on the potential impact of this low response rate on the results of study?

True, the response rate was low. This was mainly due to the difficulty in contacting parents of children in the boarding school section, if they did not have a telephone. We discussed this as an important study limitation, and stated that we do not have information on whether they were more likely to come from rural areas or the city, which areas are associated with a lower or higher asthma risk, respectively. We have included this last point in the discussion.

“Finally, a large number of potential participants were not included in this study, mostly because they were in the boarding school section and were unable to contact their parents/guardians in time to attend the parents’ meeting (to provide consent). We do not have information on whether these parents were more likely to reside in rural areas or the city, where the asthma risk is lower or higher than the town where this study was conducted, respectively.”

Also, what was the difference in the response rate between cases and controls?

This is an important point, but we do not have information on response rates of cases separate from controls because the majority of potential study participants were lost at the pre-screening stage, before we could ascertain whether they were true cases or controls.

Analytic strategy:The statistical analysis section should more coherently present confounders and mediators identified among the measured variables, statistical models used, and measures of strength and precision used to describe results. Below are a few specific comments that should either be clarified or considered through the conduct of further analysis:1) One of the major concerns is that the sample in the study is drawn from 50+ schools and thus prone to clustering effects. There is no indication in the data analysis of whether this issue was considered. We strongly recommend that the authors address and justify what they have done. If they cannot, please redo the analyses.

Thank you for this observation. We initially had not considered clustering in the analysis, since several similar previous asthma studies, particularly the ISAAC studies, have not found any significant clustering effects (Asher MI, Anderson HR, Stewart AW, Crane J, Ait-Khaled N, Anabwani G, Anderson HR, Beasley R, Björkstén B, Burr ML, Clayton TO, Crane J, Ellwood P, Keil U, Lai CKW, Mallol J, Martinez FD, Mitchell EA, Montefort S, Pearce N, Robertson CF, Shah JR, Sibbald B, Strachan DP, Weiland SK, Williams HC (ISAAC Steering Committee). Worldwide variations in the prevalence of asthma symptoms: International Study of Asthma and Allergies in Childhood (ISAAC). Eur Respir J 1998; 12: 315-335)

In accordance with the reviewers’ recommendation, we have repeated the analysis, using a random effects logistic regression analysis to allow for clustering by school. As with the ISAAC studies, allowing for clustering had minimal effects on the results, but we present these results for the benefit of some of your readers who prefer this type of analysis. We have changed the data management and statistical analysis section to reflect this change. We have also changed the adjusted odds ratios in all results tables and in the text of the Results section.

“We conducted initial logistic regression analyses for each exposure of interest, adjusted for age and sex, with random effects to allow for any clustering by school. We then ran the adjusted random effects logistic regression model (i.e. adjusted for the above factors in addition to age and sex) for each exposure of interest, and checked for collinearity by comparing the standard error of the exposure coefficient in the adjusted model and in the basic model(Greenland, Daniel and Pearce, 2016; Greenland and Pearce, 2015). We did not find any problems of collinearity, so we reported the findings for the adjusted model for each exposure.”

2) The authors state that they have assessed combined effects of risk factors on additive and multiplicative scales. They have also interpreted the results given in tables under the combined effect Results section as additive or multiplicative. However, it can be argued that the authors do not present sufficient results to make such interpretations. What were the measures and associated confidence intervals based on and which combined effect on additive or multiplicative scales were assessed? P-values are not enough to estimate strength and precision of interaction results. If the authors have not used any such measures, it is recommended to just present stratum specific effects of these exposures and refrain from making any claims about interaction. Please highlight these in the limitations.

We thank the reviewers for this assessment and we have followed their advice to present only the stratum specific effects of the exposures, and have refrained from making any claims about interaction. Because we have no longer included interactions, we have not needed to include any discussion of interactions in the limitations section.

We have removed p-values from Tables of results 4-7, and from Supplementary files 2 and 3.

3) While we appreciate that the causal pathway factors were not adjusted in the models, there is no explanation of what those factors were. Was there any technique used (e.g., causal graphs) to identify them?

Thank you for raising this important issue. We did not identify any mediators on the causal pathway. We have changed the text in the statistical analysis section to read as seen below.

“We did not identify any factors which were likely to be on the causal pathways (and therefore were not potential confounders and should not be adjusted for).”

4) The paper would benefit from an explanation for the choice of socioeconomic indicator used in the final models (Why father's education? Were the continuous variables used as linear functions in the models? Was the potential for non-linearity tested?

We collected data on both father’s and mother’s highest education level attained, as a proxy indicator for socioeconomic status. This was based on our experience conducting research in this setting, and from findings from a large study conducted in Dar es Salaam, Tanzania and Cape Town, South Africa, which found that father’s and mother’s education were significantly associated with socioeconomic status sum score (Aarø LE, Flisher AJ, Kaaya S, Onya H, Namisi FS, Wubs A. Parental education as an indicator of socioeconomic status: improving quality of data by requiring consistency across measurement occasions. *Scand J Public Health*. 2009 Jun;37 Suppl 2:16-27. doi: 10.1177/1403494808086917).

But since father’s and mother’s education were strongly associated, we chose to include only father’s education to avoid collinearity. Fathers are the breadwinners in most households in Uganda. We have included this point in the Materials and methods section.

“Previous studies in this setting have found that father’s and mother’s education are significantly associated with socioeconomic status.”

Several exposures were tested against a single outcome. Did the authors correct for multiple comparisons? If not, why?

We did not correct for multiple comparisons, because all of the tested exposures were selected because there was good prior evidence that they were likely to be risk factors for asthma. Thus, we assessed the findings for each exposure individually, in light of prior knowledge and evidence.

5) The 95% CIs were estimated in the analysis. However, this is not stated in the analysis section.

We thank the reviewer for reminding us to state explicitly in the statistical analysis section that we estimated 95% CIs. We have done this and now the statement reads as below.

“The odds ratio was the main outcome measure (as is appropriate for a case-control study), and we also estimated the 95% CIs. We identified age and sex as a priori confounders, and all analyses were adjusted for these variables. We identified area of residence at time of birth and father’s education as potential confounders; we did not also adjust for area where the child spent most of the first five years or mother’s education, since these were strongly associated with the above two factors, and therefore would have introduced collinearity.”

Also, please clarify the relevance of p-values (again not stated in the Materials and methods section) in the results tables. There is a strong argument in the literature that estimates and CIs, not p-values, convey the strength and precision of results (please refer to Nature comment: "Scientists rise up against statistical significance", 20 March 2019)

In line with the reviewer’s recommendation, we have now excluded p-values from all results tables. However, we note that your decision letter states that *“please report exact p-values wherever possible alongside the summary statistics and 95% confidence intervals”*, which is what we had done originally. We are happy to follow your advice whether to include p-values or not.

All results tables now do not include p-values.

6) Table 1 shows missingness (ranging from N=0 to N=420) related to several variables. How did the authors deal with missing values? Were imputation and sensitivity analysis considered? Any comments related to potential bias?

We thank the reviewer for raising this important issue and agree that this needs further clarification. We have repeated the analysis using the ‘complete case’ data set, by dropping 30 participants from the analysis who had missing values in the four key variables. This has not changed the study results. We have reflected this change in the number of participants in the flow diagram, results tables and in the re-written statistical section, as shown below.

“The key variables adjusted for (see below) were age, sex, area of residence at time of birth and father’s education. Each of these variables (except sex) had small numbers of missing values (the largest number was 29 for father’s education). We therefore created a ‘complete case’ data set, comprising the 555 cases and 1,115 controls which had no missing values for these variables. This ‘complete case’ data set was used for all analyses.”

[Editors' note: further revisions were requested prior to acceptance, as described below.]

The manuscript has been improved but there are some remaining issues that need to be addressed before acceptance, as outlined below:1) In subsection “Current features of asthma cases versus controls”, the authors say that there was weak evidence for an inverse association between asthma and infection with any helminth's species. Based on the odds ratio and confidence interval presented, we suggest the authors to further elaborate this conclusion.

Thank you for this observation. We agree this point needed further clarification. We have made the following changes.

“There was weak evidence for an inverse association between asthma and infection with any helminths species [0.75 (0.53-1.07)] (Table 2); overall this was not statistically significant as shown by the fact that the confidence interval included the null value of 1, but it was statistically significant for *T. trichiura* [0.33 (0.13-0.89)] (Supplementary file 1a).”

2) Please check the order of sections in the manuscript, table numbers, titles etc.

Thank you for this comment. We structured the sections of our article as follows: Abstract, Introduction, Results, Discussion, Materials and methods, References, Tables of results and Figure legends. We are willing to change this order to any other that you advise us to. We noticed that one Table was in the wrong place. We have now moved that Table from the Materials and methods section to the ‘Tables section’ and have now labelled it as “Table 8”, and have given it a title. In additional, we have changed the names for the supplementary tables to supplementary files.

3) Please check for duplicates in reference, e.g., 53 and 56.

Thank you for that observation. We have deleted reference 56 and checked that there are no more duplicate references.